# Highly efficient multi-resonance thermally activated delayed fluorescence material toward a BT.2020 deep-blue emitter

Junki Ochi [1], Yuki Yamasaki[2], Kojiro Tanaka[1], Yasuhiro Kondo[3], Kohei Isayama[3], Susumu Oda[4], Masakazu Kondo[5] & Takuji Hatakeyama [1] ✉

An ultrapure deep-blue multi-resonance-induced thermally activated delayed fluorescence material (DOB2-DABNA-A) is designed and synthesized. Benefiting from a fully resonating extended helical π-conjugated system, this compound has a small $\Delta E_{ST}$ value of 3.6 meV and sufficient spin−orbit coupling to exhibit a high-rate constant for reverse intersystem crossing ($k_{RISC} = 1.1 \times 10^6 \, s^{-1}$). Furthermore, an organic light-emitting diode employing DOB2-DABNA-A as an emitter is fabricated; it exhibits ultrapure deep-blue emission at 452 nm with a small full width at half maximum of 24 nm, corresponding to Commission Internationale de l'Éclairage (CIE) coordinates of (0.145, 0.049). The high $k_{RISC}$ value reduces the efficiency roll-off, resulting in a high external quantum efficiency (EQE) of 21.6% at 1000 cd m$^{-2}$.

Displays based on organic light-emitting diodes (OLEDs) have widely been utilized in commercial products such as TVs and smartphones[1]. The BT.2020 standard defined by the International Telecommunication Union Radiocommunication Sector (ITU-R) specifies the current requirements for ultrahigh definition (UHD) displays. This standard covers diverse aspects related to UHD displays. Notably, as the benchmark for designing emitting materials, red, green, and blue primaries are specified using Commission Internationale de l'Éclairage (CIE) chromaticity coordinates of (0.708, 0.292), (0.170, 0.797), and (0.131, 0.046), respectively; 99.9% of all natural colors can be reproduced by combining those primaries. Thus, emitters that satisfy the BT.2020 requirements are in high demand.

Thermally activated delayed fluorescence (TADF) materials have attracted increasing attention as efficient emitters for fabricating OLEDs. The main advantage of TADF-based OLEDs is that almost 100% internal quantum efficiency can be achieved without employing precious metals[2–8]. In basic terms, the best way to achieve high TADF efficiency involves the spatial separation of the highest occupied molecular orbital (HOMO) and lowest unoccupied molecular orbital (LUMO) by connecting the donor and acceptor groups[3–7]. However, such donor−acceptor-type designs have the significant drawback of a broad emission band caused by large structural relaxation in the excited state. Consequently, a color filter or microcavity is required to cut off the margin region and enhance the color purity for application in the development of OLED displays, resulting in a significant energy loss.

To overcome this drawback, we previously developed an alternative approach for designing TADF materials based on the multi-resonance (MR) effect[9–16]. In MR-TADF materials, the complementary resonance effects of boron[17–23] and nitrogen atoms realize the alternate localization of HOMOs and LUMOs at different carbon atoms on the same benzene ring. The resulting HOMO−LUMO separation successfully suppresses the structural relaxation and vibronic coupling, enabling a high photoluminescence quantum yield (PLQY) and high color purity. Since the development of the pure blue MR-TADF material, DABNA-1[9], extensive research has been conducted in the field of MR-TADF materials[24–40]. In particular, π-extension has been effectively used to increase the reverse intersystem crossing rate constant ($k_{RISC}$), which is important for avoiding a severe efficiency roll-off at high voltage[30]. However, owing to the spectral redshift associated with π-extension, a material with a high $k_{RISC}$ value and color purity adequate for a BT.2020 blue emitter has not been realized to date[41–50].

[1]Department of Chemistry, Graduate School of Science, Kyoto University, Sakyo-ku, Kyoto 606-8502, Japan. [2]Department of Chemistry, Graduate School of Science and Technology, Kwansei Gakuin University, 2-1 Gakuen, Sanda, Hyogo 669-1337, Japan. [3]SK JNC Japan Co., Ltd., 5-1 Goi Kaigan, Ichihara, Chiba 290-8551, Japan. [4]Department of Applied Chemistry, Graduate School of Science and Engineering, Toyo University, 2100 Kujirai, Kawagoe, Saitama 350-8585, Japan. [5]JNC Co., 5-1 Goi Kaigan, Ichihara, Chiba 290-8551, Japan. ✉e-mail: hatake@kuchem.kyoto-u.ac.jp

To design highly efficient pure deep-blue MR-TADF emitters, a computational investigation using double hybrid time-dependent density functional theory calculations[51] was conducted for model-A and model-B; here, efficient π-extension can be achieved by merging DABNA-1 with two DOBNA[52] units (Fig. 1 and Supplementary Fig. 3). Because the DOBNA unit possesses the highest $S_1$ energy (3.43 eV) in MR-TADF materials, it was selected to keep deep-blue emission even after merging with other π-conjugated frameworks. First, the molecular geometries in the $S_1$ state were optimized to evaluate the fluorescence process. The degree of structural distortion was different for the two model compounds. In model-A, two oxygen atoms were closely placed to cause steric repulsion. Consequently, the dihedral angle between benzene rings *a* and *b* was larger in model-A (34.09°) than in model-B (17.18°). The moderate helicity allowed the molecular orbitals to delocalize over the whole structure in both models. The enlarged π-conjugation decreased both the $\Delta E_{S1-T1}$ (23 meV for model-A, 80 meV for model-B, and 239 meV for DABNA-1) and $\Delta E_{S1-T2}$ (231 meV for model-A, 367 meV for model-B, and 586 meV for DABNA-1) values. The spin-orbit coupling (SOC) matrix elements ($\langle S_n | \hat{H}_{SOC} | T_n \rangle$) calculated at an M062X/TZP level of theory indicate that model-A has a large SOC ($S_1-T_1$: 0.059 cm⁻¹, $S_1-T_2$: 0.895 cm⁻¹) compared to model-B ($S_1-T_1$: 0.035 cm⁻¹, $S_1-T_2$: 0.206 cm⁻¹), benefiting from the more helical structure[53]. Although the contribution of $T_2$ to the RISC process is expected to be very small due to the large $\Delta E_{S1-T2}$ values, the small $\Delta E_{S1-T1}$ and the larger SOC for $S_1-T_1$ will improve the TADF properties. Moreover, the transition energy between the $S_0$ and $S_1$ states was predicted to be

higher in model-A (2.682 eV, 462 nm) than in model-B (2.529 eV, 490 nm). These results indicate that model-A is a promising framework for ultrapure blue emitters with a high $k_{RISC}$ value, favorable for the fabrication of OLED devices[53–59].

## Results

### Molecular design and synthesis

Motivated by the computational results described above, a new MR-TADF blue emitter, DOB2-DABNA-A-NP, was designed based on model-A. A diphenylamine group was introduced to the central benzene ring to enhance the nucleophilicity. The synthesis of DOB2-DABNA-A-NP is shown in Fig. 2: Buchwald−Hartwig coupling between 3,5-dichloro-*N,N*-diphenylaniline and DOBNA-NHPh afforded intermediate **1**. In the presence of boron triiodide and 2,6-di-*tert*-butylpyridine, the following borylation of **1** consisting of intermolecular and intramolecular electrophilic C−H borylation[60–62] smoothly took place at room temperature. Since we observed over borylation at the peripheral positions (not identified), the crude product was treated with acetic acid to remove the boryl groups to afford DOB2-DABNA-A-NP in 42% yield. This borylation method, known as one-shot borylation[63–66], was also applicable for synthesizing DOB2-DABNA-A, which possesses a *tert*-butyl group instead of a diphenylamine group to suppress unfavorable intermolecular borylation at the central benzene ring. In addition, an *m*-xylyl group (denoted Xyl) was adopted as an *N*-attached aryl group to suppress the over borylation. As a result, DOB2-DABNA-A was selectively obtained in 40% yield without any treatment with acetic acid. As a reference compound based on model-B, we also synthesized

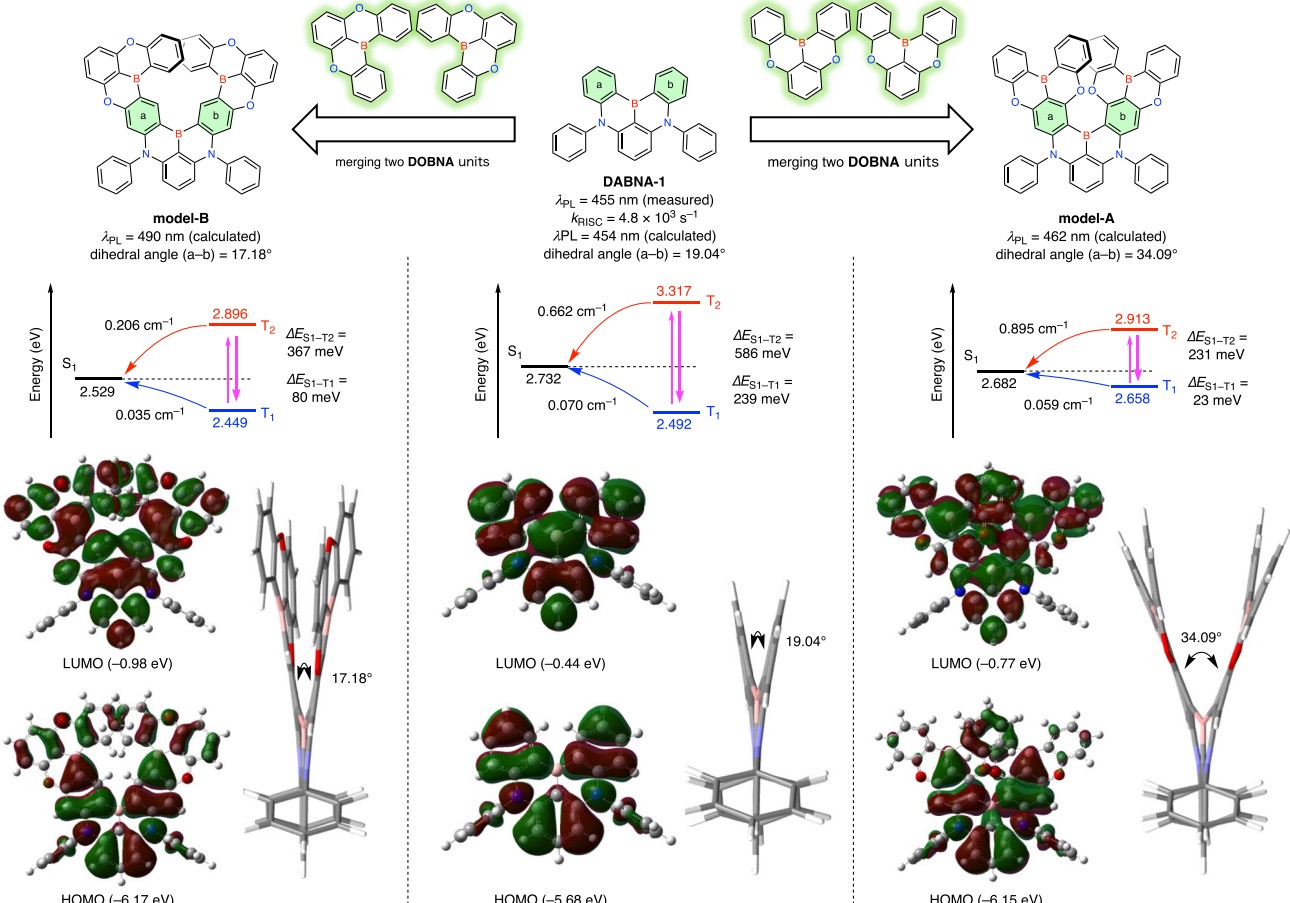

**Fig. 1 | Energy-level diagrams, frontier molecular orbitals (isovalue = 0.01), and side views of DABNA-1, model-A, and model-B with S1 geometry.** Transition energies for $S_1$, $T_1$, and $T_2$ were calculated at the TDA-B2PLYP (cx = 0.40, cc = 0.23)/ cc-PVDZ//M062X/6-31 G(d) levels of theory. SOC matrix elements were calculated at the M062X/TZP//M062X/6-31 G(d) level of theory.

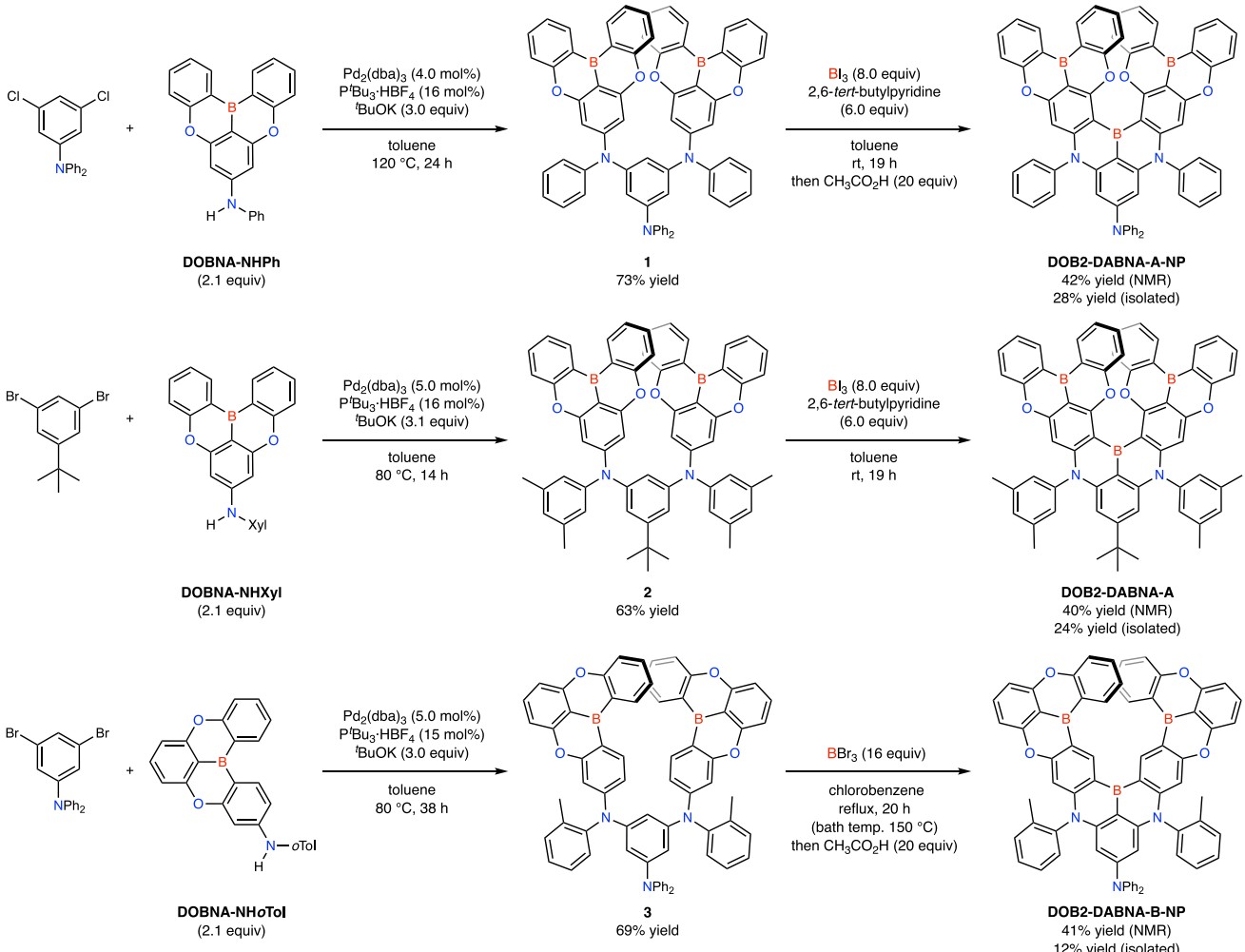

**Fig. 2 | Synthesis of DOB2-DABNA-A-NP, DOB2-DABNA-A, and DOB2-DABNA-B-NP.** Buchwald–Hartwig coupling and the following one-shot borylation afforded the target compounds.

DOB2-DABNA-B-NP possessing a diphenylamine group. To suppress undesired intramolecular borylation on the *N*-attached aryl groups, *o*-tolyl groups were used instead of phenyl and *m*-xylyl groups. Since the borylation using boron triiodide and 2,6-di-*tert*-butylpyridine gave a complex mixture, we chose boron tribromide as a milder borylating reagent. As a result, the reaction took place under reflux condition of chlorobenzene to give DOB2-DABNA-B-NP in 41% yield upon removal of the borylated groups at the peripheral positions by treatment with acetic acid.

**Photoluminescence properties**

The photophysical properties of DOB2-DABNA derivatives measured in 1 wt%-doped poly(methyl methacrylate) (PMMA) films are summarized in Fig. 3, Table 1, Supplementary Fig. 4, and Supplementary Table 2. For comparison, the parent framework, DABNA-1, is also listed in Table 1. DOB2-DABNA-A exhibited an ultrapure deep-blue emission at 451 nm, shorter than that of DABNA-1 (455 nm). The full width at half maximums (FWHMs) and PLQY values were determined as 27 nm (170 meV) and 92% (Fig. 3a), respectively. Based on the peak top of the fluorescence and phosphorescence spectra at 77 K, the $S_1$ and $T_1$ energies were determined as 2.745 and 2.742 eV, respectively. The $S_1$ energy was not so different from DABNA-1 (2.71 eV) because the two nitrogen or two boron atoms in the *meta*-position are not effectively conjugated and change the orbital energy little, which is the general trend in MR-TADF materials[10]. In contrast, the $T_1$ energy of DOB2-DABNA-A was much higher than that of DABNA-1 (2.54 eV)

(Supplementary Fig. 5). This originates from the significant decrease in $\Delta E_{ST}$ value (3.6 meV for DOB2-DABNA-A and 170 meV for DABNA-1, respectively), which can be attributed to the charge delocalization through the π-extension[30]. Next, the transient decay spectra were measured to evaluate the TADF properties (Fig. 3b). The spectra of DOB2-DABNA-A include two components corresponding to a prompt lifetime of 6.21 ns and a delayed lifetime of 1.55 μs. Based on the determined quantum yields and emission lifetimes, the rate constants for fluorescence ($k_F$), internal conversion ($k_{IC}$), and intersystem crossing ($k_{ISC}$) and the $k_{RISC}$ values were calculated according to a literature method (Fig. 3b)[67,68]. Notably, the $k_{RISC}$ value of DOB2-DABNA-A is $1.1 \times 10^6$ s$^{-1}$, higher than those of DABNA-1 ($4.8 \times 10^3$ s$^{-1}$), corresponding to a smaller $\Delta E_{ST}$ value.

To evaluate the effect of substituents at the central phenyl ring, the photophysical properties of DOB2-DABNA-A and DOB2-DABNA-A-NP were compared. DOB2-DABNA-A-NP showed slightly shorter $\lambda_{max}$ (446 nm, Fig. 3c) and similar $k_{RISC}$ ($1.2 \times 10^6$ s$^{-1}$, Fig. 3d). These data indicate that the diphenylamine group affects excitation energy but does not significantly affect RISC process. To verify this, we conducted NTO calculations for DOB2-DABNA-A and DOB2-DABNA-A-NP and confirmed that substituents at the central phenyl ring had little contribution to the $S_0$–$S_1$ transition (Supplementary Fig. 2). DOB2-DABNA-A-NP showed lower ($\Phi = 0.76$) PLQY and broader emission spectrum (FWHM = 29 nm) than DOB2-DABNA-A, probably due to aggregation in the PMMA film. Next, DOB2-DABNA-A-NP and DOB2-DABNA-B-NP were compared to investigate the effect of the

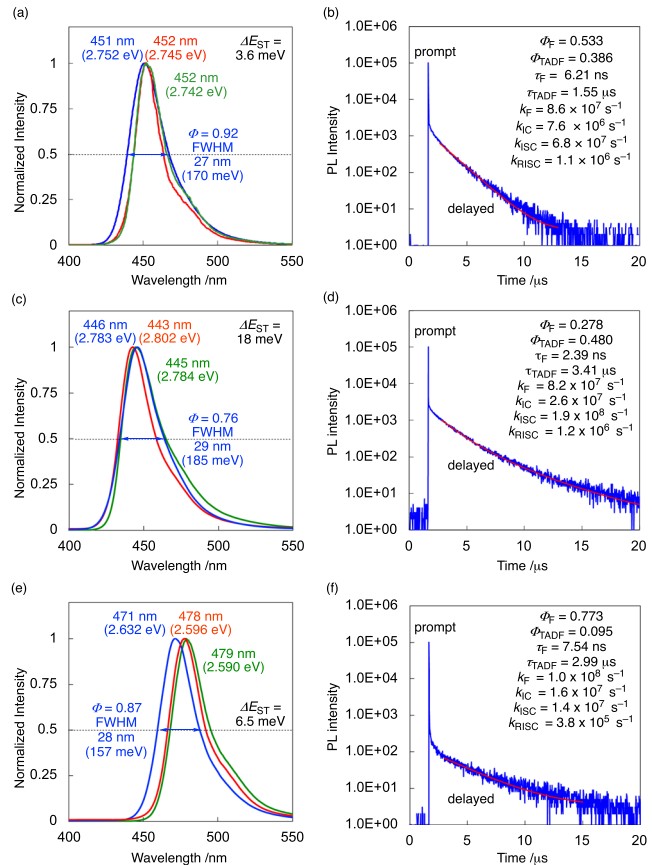

**Fig. 3 | Photophysical properties of DOB2-DABNA-A (a,b), DOB2-DABNA-A-NP (c,d), and DOB2-DABNA-B-NP (e,f) in poly(methyl methacrylate) (PMMA; 1 wt %-doped films).** (a,c,e) Photoluminescence spectra at 300 K (blue) and 77 K with (green) and without (red) a delay time of 25 ms. (b,d,f) Transient photoluminescence decay curves at 300 K and their relevant parameters. The red curves represent the single exponential fitting data (background = 1–3).

backbone molecular structure. Consequently, DOB2-DABNA-B-NP exhibited longer wavelength emission (471 nm) than DOB2-DABNA-A-NP (Fig. 3e), which is in good agreement with the computational prediction in Fig. 1. Moreover, benefiting from its helical structure, the $k_{RISC}$ value was three times larger in DOB2-DABNA-A-NP ($1.2 \times 10^6 \, \text{s}^{-1}$) than in DOB2-DABNA-B-NP ($3.8 \times 10^5 \, \text{s}^{-1}$, Fig. 3f). The higher $k_{RISC}$ in DOB2-DABNA-A derivatives than DOB2-DABNA-B-NP could be reproduced by semi-classical Marcus equation using computational and experimental parameters (Supplementary Table 3). Transient spectra of all three compounds at different temperatures showed that the TADF component vs. the prompt one decreased at lower temperatures (Supplementary Fig. 9). From the Arrhenius plots of $k_{RISC}T^{0.5}$ and $k_{ISC}T^{0.5}$ versus $1/T$, the activation energies for the RISC ($\Delta E_a^{RISC}$) and ISC ($\Delta E_a^{ISC}$) processes were estimated to be 64 and 27 meV (for DOB2-DABNA-A), 70 and 26 meV (for DOB2-DABNA-A-NP), and 43 and 52 meV (for DOB2-DABNA-B-NP), respectively. The reversal of the activation energies in DOB2-DABNA-B-NP can be attributed to the dark state in equilibrium with the $T_1$ state, which may reduce population of the $T_1$ state at higher temperature to decrease $k_{RISC}$ and the slope ($\Delta E_a^{RISC}$) of the fitting line.

Considering their small FWHM and high PLQY values, DOB2-DABNA-A and DOB2-DABNA-B-NP should be promising candidates for application as MR-TADF emitters for OLED devices. To evaluate the potential as OLED emitters, their photophysical properties in the 1 wt%-doped DOBNA-Tol host material[64] were measured (Supplementary Fig. 6). The fluorescence peaks were at the same position as the PMMA films. The FWHM values of DOB2-DABNA-A and DOB2-

DABNA-B-NP in DOBNA-Tol were 24 nm and 23 nm, which was smaller than those in PMMA (27 nm and 28 nm). This can be attributed to aggregation of DOB2-DABNA-A in the 1 wt%-doped PMMA films, as also suggested by the decreased FWHM value of 0.1 wt %-doped PMMA film (23 nm, Supplementary Fig. 7). Furthermore, the $k_{RISC}$ values of DOBNA-Tol neat film ($8.8 \times 10^4 \, \text{s}^{-1}$) and the 1 wt %-doped DOBNA-Tol films ($6.0 \times 10^5 \, \text{s}^{-1}$ for DOB2-DABNA-A and $2.0 \times 10^5 \, \text{s}^{-1}$ for DOB2-DABNA-B-NP) were smaller than the 1 wt %-doped PMMA films ($1.1 \times 10^6 \, \text{s}^{-1}$ and $3.8 \times 10^5 \, \text{s}^{-1}$) (Supplementary Figs. 6 and 8). This allows us to rule out the possibility of a sensitizing effect of DOBNA-Tol. Note that the unusual $\lambda_{max}$ shift between 300 K and 77 K observed for DOB2-DABNA-B-NP in the PMMA film (Fig. 3e) was not reproduced in the DOBNA-Tol film.

**OLED performances**
Finally, devices with the following structure were fabricated: indium tin oxide (ITO, 50 nm); N,N'-di(1-naphthyl)-N,N'-diphenyl-(1,1'-biphenyl)-4,4'-diamine (NPD, 40 nm); tris(4-carbazolyl-9-ylphenyl)amine (TCTA, 15 nm); 1,3-bis(N-carbazolyl)benzene (mCP, 15 nm); 1 wt% emitter (DOB2-DABNA-A or DOB2-DABNA-B-NP) and 99 wt% DOBNA-Tol[64] (20 nm); 3,4-di(9H-carbazol-9-yl)benzonitrile (3,4-2CzBN[69], 10 nm); 2,7-bis(2,2'-bipyridine-5-yl)triphenylene (BPy-TP2[70], 20 nm); LiF (1 nm); and Al (100 nm). The electroluminescence characteristics, ionization potentials, and electron affinities of the fabricated devices are shown in Fig. 4 and Supplementary Figs. 13, 14. The device employing DOB2-DABNA-A exhibited an ultrapure deep-blue emission at 452 nm with an FWHM of 24 nm (146 meV) and corresponding CIE coordinates of (0.145, 0.049). These values are very similar to the blue-color space parameter (0.131, 0.046) required by the BT.2020 standard for UHD displays with a wide color scope (Fig. 4b, c). The device employing DOB2-DABNA-B-NP also showed narrowband blue emission. However, the EL spectrum peak (471 nm) was longer than DOB2-DABNA-A, and the CIE coordinates of (0.117, 0.127) went away from the pure blue region. The FWHM values of the EL spectra were 23–24 nm, agreeing with the PL spectra in DOBNA-Tol film (Supplementary Fig. 6). The fabricated device can be driven using voltages as low as 3.2–3.4 V (Fig. 4d) and shows a maximum external quantum efficiency (EQE) of 24.1–29.1% (Fig. 4e), comparable to previous MR-TADF material-based devices[9–15]. Especially, the DOB2-DABNA-A-based device showed the best EQE value at the practical luminance among the previously reported OLEDs with CIE$_y$ ≤ 0.05 (Fig. 4f and Supplementary Table 4). The EQE value was maintained within 21.6% at 1000 cd m$^{-2}$, benefiting from a suppressed efficiency roll-off (10.4%). This can be attributed to the high $k_{RISC}$ value (Fig. 4g and Supplementary Table 5) which is critical to suppress the quenching pathways from the triplet state at high current density. Moreover, the half-lifetime (LT$_{50}$) of the DOB2-DABNA-A-based device with an initial luminance of 100 cd m$^{-2}$ was 52 h, which is over ten times longer than the previous deep-blue MR-TADF emitter-based OLEDs (Supplementary Fig. 14 and Supplementary Table 4)[41,43,44,50].

**Discussion**
In conclusion, we synthesized an ultrapure deep-blue MR-TADF material, DOB2-DABNA-A, by sequential one-pot borylation reactions. The large SOC value induced by the helical structure significantly increased the $k_{RISC}$ value for DOB2-DABNA-A ($1.1 \times 10^6 \, \text{s}^{-1}$). The OLED devices fabricated with DOB2-DABNA-A as an emitter exhibited ultrapure deep-blue emissions at 452 nm with an FWHM of 24 nm and corresponding CIE coordinates of (0.145, 0.049). This almost satisfies the requirements for blue displays, as defined by BT.2020. Moreover, the device demonstrated a maximum EQE of 24.1% and an efficiency roll-off of 10.4% at 1000 cd m$^{-2}$, a record-setting value for deep-blue (CIE$_y$ ≤ 0.05) MR-TADF materials[41–50]. Thus, this work will pave the way for designing and constructing highly efficient deep-blue MR-TADF emitters for UHD displays.

**Table 1 | Photophysical properties of DOB2-DABNA-A, DOB2-DABNA-A-NP, DOB2-DABNA-B-NP, and DABNA-1 in PMMA (1 wt %-doped films)**

| Compound | $\lambda_{max}$ [nm] | FWHM [nm] | $\Phi$ [a)] | $\Phi_F$ [b)] | $\Phi_{TADF}$ [b)] | $\tau_F$ [c)] [ns] | $\tau_{TADF}$ [c)] [$\mu$s] | $k_F$ [d)] [$10^7$ s$^{-1}$] | $k_{IC}$ [d)] [$10^7$ s$^{-1}$] | $k_{ISC}$ [d)] [$10^7$ s$^{-1}$] | $k_{RISC}$ [d)] [$10^4$ s$^{-1}$] |
|---|---|---|---|---|---|---|---|---|---|---|---|
| DOB2-DABNA-A | 451 | 27 | 0.92 | 0.53 | 0.39 | 6.21 | 1.55 | 8.6 | 0.76 | 6.8 | 112 |
| DOB2-DABNA-A-NP | 446 | 29 | 0.76 | 0.28 | 0.48 | 2.39 | 3.41 | 8.2 | 2.6 | 19 | 115 |
| DOB2-DABNA-B-NP | 471 | 28 | 0.87 | 0.77 | 0.10 | 7.54 | 2.99 | 10 | 1.6 | 1.4 | 38 |
| DABNA-1 | 455 | 29 | 0.82 | 0.73 | 0.09 | 11.5 | 235 | 6.4 | 1.4 | 0.94 | 0.48 |

a)Absolute photoluminescence quantum yield

b)Fluorescent and TADF components determined from the total $\Phi$ and contribution of the integrated area of each component in the transient spectra to the total integrated area

c)Lifetimes calculated from fluorescence decay

d)Rate constants for fluorescence ($k_F$), internal conversion from $S_1$ to $S_0$ ($k_{IC}$), intersystem crossing from $S_1$ to $T_1$ ($k_{ISC}$), and reverse intersystem crossing from $T_1$ to $S_1$ ($k_{RISC}$) were calculated from $\Phi$, $\Phi_F$, $\Phi_{TADF}$, $\tau_F$, and $\tau_{TADF}$ according to Adachi's method[67,68].

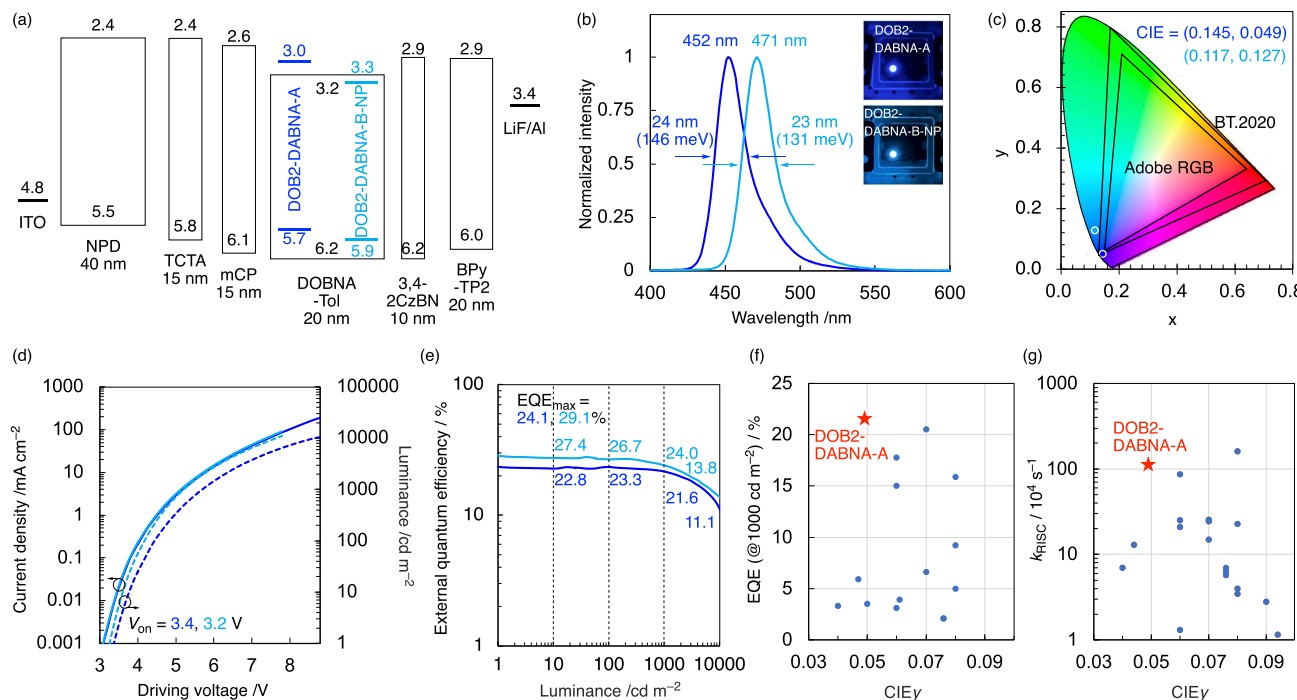

**Fig. 4 | Characteristics of fabricated OLED device using DOB2-DABNA-A (blue) and DOB2-DABNA-B-NP (light blue) as an emitter. a** Device structure, ionization potentials ($I_p$), and electron affinities ($E_a$; in eV) for each component. The $I_p$ and $E_a$ of emitters were estimated from those of DABNA-1 and their HOMO/LUMO energy levels (Supplementary Table 6). **b** Normalized EL spectra of the devices in operation. Inset: electroluminescence of the device. **c** Commission Internationale de l'Éclairage (CIE) (x,y) coordinates. **d** Current density (solid) and luminance (dashed) vs. driving voltage. **e** EQE vs. luminance. **f** Comparison of EQE values at 1000 cd m$^{-2}$ for reported deep-blue OLEDs with CIE$_y$ < 0.1. **g** Comparison of $k_{RISC}$ values at 1000 cd m$^{-2}$ for reported deep-blue OLEDs with CIE$_y$ < 0.1.

## Methods

### General procedure

All the reactions dealing with air- or moisture-sensitive compounds were carried out in a dry reaction vessel (small scale, a Schlenk flask; large scale, a three-necked round bottomed flask) under a positive pressure of nitrogen. Air- and moisture-sensitive liquids and solutions were transferred via a syringe or a Teflon® cannula. Analytical thin-layer chromatography (TLC) was performed on glass plates coated with 0.25 mm 230–400 mesh silica gel containing a fluorescent indicator (Merck, #1.05715.0009). TLC plates were visualized by exposure to ultraviolet light (254 nm or 365 nm). Organic solutions were concentrated by rotary evaporation at *ca.* 10–50 mmHg. Flash column chromatography was performed on Merck silica gel 60 (spherical, neutral, 140–325 mesh) and Kanto Chemical silica gel 60 N (spherical, neutral, 40–50 μm). Proton nuclear magnetic resonance (¹H NMR), carbon nuclear magnetic resonance (¹³C NMR), and boron nuclear magnetic resonance (¹¹B NMR) spectra were recorded on JEOL ECA500 (500 MHz) NMR spectrometers. Proton chemical shift values are reported in parts per million (ppm, δ scale) downfield from tetramethylsilane and are referenced to the tetramethylsilane (δ 0). ¹³C NMR spectra were recorded at 126 MHz: carbon chemical shift values are reported in parts per million (ppm, δ scale) downfield from tetramethylsilane, and are referenced to the carbon resonance of tetramethylsilane (δ 0) or CDCl₃ (δ 77.0). ¹¹B NMR spectra were recorded at 160 MHz: boron chemical shift values are reported in parts per million (ppm, δ scale) and are referenced to the external standard boron signal of BF₃·Et₂O (δ 0). Data are presented as: chemical shift, multiplicity (s = singlet, d = doublet, t = triplet, m = multiplet and/or multiplet resonances, br = broad), coupling constant in hertz (Hz), signal area integration in natural numbers, and assignment (*italic*). IR spectra were recorded on an ATR-FTIR spectrometer (FT/IR-4200, JASCO or IRAffi-nity-1S, Shimadzu). Characteristic IR absorptions are reported in cm⁻¹. Melting points were recorded on a Fisher-Johns 12-144-1Q melting point apparatus (according to the limitations of the apparatus, the

compounds which did not melt up to 300 °C are presented as ">300 °C"). High-resolution mass spectra (HRMS) were obtained by the matrix assisted laser desorption/ionization (MALDI) method with a JEOL SpiralTOF instrument. Purity of isolated compounds was determined by $^1$H NMR analyses or HPLC analysis on a JASCO UV-2070 Plus instrument equipped with a reversed-phase C18 column (Mightysil RP-18 GP, Kanto Chemical Co., Inc., 4.6 mm × 100 mm i.d.). For purification by HPLC, reversed-phase C18 column (Mightysil RP-18 GP, Kanto Chemical Co., Inc., 20 mm × 250 mm i.d.)

### Materials
Materials were purchased from Wako Pure Chemical Industries, Ltd. (Wako), Tokyo Chemical Industry Co., Ltd., Aldrich Inc., and other commercial suppliers, and were used after appropriate purification, unless otherwise noted. Florisil (100–200 mesh) was purchased from Kanto Chemical Co., Inc. (Kanto).

### Solvent
Anhydrous solvents were purchased from above-described suppliers and/or dried over Molecular Sieves 4 A and degassed before use. Water content of the solvent was determined with a Karl Fischer moisture titrator (AQ-2200, Hiranuma Sangyo Co., Ltd.) to be less than 20 ppm.

### Synthesis and characterization
The experimental details on synthesis and characterization are presented in the Supplementary Information. The NMR charts are shown in Supplementary Figs. 15–40.

### Computational methods
All calculations were performed with the Gaussian 16 (Revision B.01), ADF2021, and PySCF packages unless otherwise noted. The DFT method was employed using the B3LYP or M062X hybrid functional. The structures were optimized using the 6–31 G(d) basis set. Calculations based on the time-dependent density functional theory (TD-DFT) were conducted at the B3LYP/6-31 G(d) or M062X/TZP levels. Further information is provided in the Supplementary Information.

### Measurement of absorption and emission characteristics
UV-visible absorption spectra were measured using a V-560 UV-visible spectrometer (JASCO) at 298 K. Photoluminescence (PL) spectra were measured using an F-7000 spectrometer (Hitachi High-Tech) at 77 and 298 K. Furthermore, the absolute PL quantum yields were measured using C9920-02G spectrometers (Hamamatsu Photonics). The PL decays were measured using a C11367 spectrometer (298 K, Hamamatsu Photonics) and were then fitted using a single exponential function to determine the lifetimes of prompt and delayed fluorescence. The measurement temperature was controlled by CoolSpeK (UNISOKU) for Arrhenius plot.

### Device fabrication and measurement of electroluminescence characteristics
OLEDs were fabricated on glass substrates coated with a patterned transparent ITO conductive layer. The substrates were treated with 300 W oxygen plasma. The pressure during the vacuum evaporation was $5.0 \times 10^{-4}$ Pa, and the film thickness was controlled using a calibrated quartz crystal microbalance during deposition. After all layers were deposited, the OLED test modules were encapsulated with a capping glass in an evaporation chamber filled with nitrogen. The OLED characteristics of all fabricated devices were evaluated at 298 K in an air atmosphere using a voltage–current–luminance measuring system, comprising a source meter (Keithley 2400) and a spectral radiance meter (Topcon SR-3AR). The EQE was calculated using the EL spectrum, assuming that the light-emitting surface was a perfect diffusion surface; all radiance elements from every angle were added up and inputted into the formula to obtain the EQE.

## Data availability
The data supporting the main findings of this work are available within the paper and its Supplementary Information, or available from the corresponding author upon request.

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

## Acknowledgements

This study was supported by JST CREST (grant number JPMJCR22B3 for T.H.) and JSPS KAKENHI (grant numbers 20H05863 and 21H02019 for T.H.). T.H. acknowledges the Asahi Glass Foundation for financial support. The authors thank Mr. Ryosuke Kawasumi (SK JNC Japan Co., Ltd.) and Dr. Takeshi Matsushita (JNC Co.) for their experimental support and valuable input.

## Author contributions

T.H. conceived and supervised the project. J.O., Y.Y., K.T., and K.I. performed the synthesis and characterization. J.O. and Y.K. performed the photophysical measurements. J.O, Y.Y., and M.K. performed the computational calculations. Y.K. fabricated the OLED devices. J.O. wrote the manuscript. T.H. and S.O revised the manuscript.

## Competing interests

The authors declare no competing interests.

## Additional information

**Supplementary information** The online version contains Supplementary Material available at https://doi.org/10.1038/s41467-024-46619-8.

