## [Peer Review File · Nature Communications]

Highly Efficient Multi-Resonance Thermally Activated Delayed Fluorescence Material Toward a BT.2020 Deep-Blue EmitterREVIEWER COMMENTS

Reviewer #1 (Remarks to the Author):

Prof. Hatakeyama and co-workers designed and synthesized a novel ultrapure deep-blue MR-TADF emitter, DOB2-DABNA-A, by merging DABBNA-1 with two BOBNA units. Benefiting from the fully resonant extended helical conjugated system, this compound has a small ΔE_{st} , high photoluminescence quantum yield, and large k_{RISC} value. Ultimately, an OLED employing DOB2-DABNA-A as an emitter exhibited ultrapure deep-blue emission at 451 nm and FWHM of 27 nm with CIE coordinates of (0.14, 0.05), which almost satisfies the BT. 2020 standard. Notably, this device realized a maximum EQE of 18% and maintained as high as 16.9% at a practical brightness of 1000 cd/m², and represents the best performance for MR-TADF emitters with $CIE_y \leq 0.05$. However, the manuscript has the following questions, requiring major revision before publication in Nature Communications.

1. Generally, increasing molecular conjugation will decrease the excited state energy level. However, compared with the parent core DABNA-1, the T1 levels of DOB2-DABNA-A and DOB2-DABNA-B are increased from 2.54 eV to 2.74 eV and 2.59 eV, respectively, and the authors need to explain this.
2. Compared with DOB2-DABNA-A, the reference compound DOB2-DABNA-B has two different substituents except for the position where BOBNA merged with DABNA-1. However, from the theoretical calculation, the diphenylamine substituent of DOB2-DABNA-B participates in the frontier orbital distribution, so DOB2-DABNA-B is not a suitable reference. It is recommended that the authors synthesize model-B reference compounds with identical substituents and study the corresponding photophysical properties.
3. The nature of the emitter is undoubtedly essential to the device's performance. Still, combined with the research work of this group and all others on MR-TADFs, the smaller efficiency roll-off of the device in this work seems to be more derived from the selection of the host material, ETL material, and blocking layer between them. The author should add relevant explanations in the text. Besides, have the authors studied the EL properties of devices based on the commonly used host materials (mCBP/TSPO1) and/or ETL (TPBi/Bphen/TmPyPB)?
4. As stated by the authors, DOB2-DABNA-B exhibited longer wavelength emissions than DOB2-DABNA-A, consistent with theoretical predictions. However, compared with DABNA-1, it is inconsistent with the theoretical prediction. Can the author give a reasonable explanation?
5. In Page 1 (line 21), it should be DOB2-DABNA-A instead of DOB2-DABNA. Besides, the efficiency roll-off of the device should be 6.1% (1.1%/18%) instead of 1.1% (line 23, page 8).

Reviewer #2 (Remarks to the Author):

The manuscript nat comm 438714_0 “Highly Efficient Multi-Resonance Thermally Activated Delayed Fluorescence Material Toward a BT.2020 Deep-Blue Emitter” describes a new MR-TADF emitters synthesis, photophysical and OLED device analysis.

Generally, the manuscript is correctly written and presents interesting results. I admire Prof. Hatakeyama's work, but unfortunately, in my opinion, this work has substantial errors that are impossible to correct without losing the main point and should be rejected.

I believe the photophysical analysis is misleading, and the OLED device is wrong (probably the NIST calibration is missing) because those materials couldn't get such high EQE values. Many authors wrongly analyse the PLQY measurement and associate all the efficiency to the DF process, which is untrue. Pure fluorescence emitter can also have PLQY 100%, but we cannot get more than 5% EQE.

Here authors present emitters with high PLQY values, but the triplet contribution is very low (table 1 - from 11% up to 0.42%). The DF/PF or ϕ_{DF}/ϕ_F values are very low <0.74 , which tells us that we even cannot use the equations mentioned by the authors because not all assumptions were fulfilled. To correctly calculate the kinetic parameters, we need strong TADF emitters with $\phi_{DF}/\phi_F > 4$ (10.1088/2050-6120/aa537e).

In that case, it would be impossible to get the EQE of the devices based on those compounds with values above 8%.

I think a broader photophysical analysis should be done, and the calibration of the photoluminescence spectrometer and OLED system should be checked.

Reviewer #3 (Remarks to the Author):

See Attachment

Ochi et *al.* have put together a comprehensive study including the synthesis of two new MR-TADF emitters, quantum chemical calculations, the photophysical characterization and OLED devices preparation. Inspired by their previous works, the authors connected their DABNA-1 and DOBNA in different fashions in order to produce two new ultrapure deep blue emitters with CIE_y ≤ 0.05 exhibiting external quantum efficiency up to nearly 17%. The study is very well conducted and I have no comment on the manuscript which is very clear.

Reviewer #4 (Remarks to the Author):

Prof. Hatakeyama et al. have presented a study on an ultrapure deep blue MR-TADF emitter, which exhibits a fast krisc of over 10^6 . The OLED demonstrates narrowband deep blue EL with CIE coordinates of (0.14, 0.05) and a mild efficiency roll-off at 1000 nits, which is an important target of OLEDs. I thus believe the current results are supportive to be a candidate for Nat. Commun. paper. However, the current data and discussions are far from enough. A major revision is needed.

1. The authors have noted that both emitters have small ΔEST values, but there is a significant difference in their krisc values. The SOC between S1 and T1 is not particularly high, so it is important to explore why DOB2-DABNA-A can achieve a krisc of over 10^6 s⁻¹. What is the contribution of higher lying triplet states. The current discussion is insufficient, and more theoretical estimations and discussions are required.

2. The authors should provide an explanation for their choice of PMMA as the dilute host. It is important to consider the potential aggregation of the MR emitters during spin-coating, as it can impact the photophysics. Please note the facts that the FWHM of DOB2-DABNA-A in PL is even boarder than that in EL spectrum, and PL of DOB2-DABNA-B at R.T is different from that at 77 K (Figure 3c), which are abnormal.

3. It is recommended to measure temperature-dependent transient PL decays and estimate the ΔEST using an Arrhenius plot based on the collected data.

4. It would be valuable to investigate whether DOBNA-Tol acts as a TADF host and plays a role as a sensitizer for the terminal emitter.

5. The measurement of device operational lifetime is crucial, particularly for deep blue OLED emitters. This parameter should be included and discussed in the study.

6. The EL performance of DOB2-DABNA-B should also be presented and discussed to provide a comprehensive analysis.

Point-by-Point Response to Reviewers' Comments

Black: Reviewer's comment Blue: Our response Red: Revision of manuscript

Reviewer #1

Prof. Hatakeyama and co-workers designed and synthesized a novel ultrapure deep-blue MR-TADF emitter, DOB2-DABNA-A, by merging DABNA-1 with two BOBNA units. Benefiting from the fully resonant extended helical conjugated system, this compound has a small ΔE_{ST} , high photoluminescence quantum yield, and large kRISC value. Ultimately, an OLED employing DOB2-DABNA-A as an emitter exhibited ultrapure deep-blue emission at 451 nm and FWHM of 27 nm with CIE coordinates of (0.14, 0.05), which almost satisfies the BT. 2020 standard. Notably, this device realized a maximum EQE of 18% and maintained as high as 16.9% at a practical brightness of 1000 cd/m², and represents the best performance for MR-TADF emitters with CIE_y ≤ 0.05. However, the manuscript has the following questions, requiring major revision before publication in Nature Communications.

Response: We thank the reviewer for their constructive comments and careful inspection of our manuscript. We have worked to address as described below thoroughly.

[Comment #1]

Generally, increasing molecular conjugation will decrease the excited state energy level. However, compared with the parent core DABNA-1, the T₁ levels of DOB2-DABNA-A and DOB2-DABNA-B are increased from 2.54 eV to 2.74 eV and 2.59 eV, respectively, and the authors need to explain this.

Response: Thank you for your important comment. The high T₁ energies of DOB2-DABNA-A and DOB2-DABNA-B-NP (renamed from DOB2-DABNA-B in the revised manuscript) originate from the comparable S₁ energies and the smaller S-T gaps. The Figure shown below (Figure r1) summarizes the S₁ and T₁ energy levels estimated from the fluorescence and phosphorescence peak tops in 1 wt% PMMA matrix. For a clear explanation, we also measured the photophysical properties of *v*-DABNA (Ref. 10, *Nat. Photonics* 2019, 13, 678.) under the same conditions. First, π -expansion does not significantly decrease the S₁ energy. This is because the two nitrogen or two boron atoms in the meta-position are not effectively conjugated, and the orbital energy changes little, which is the general trend in MR-TADF materials as exemplified by DABNA-1 (2.71 eV) and *v*-DABNA (2.66 eV). Similarly, the S₁ energies of DOB2-DABNA-A/B (2.75 and 2.60 eV) are very close to that of DABNA-1 (2.71 eV). However, the S₁ energies of DOB2-DABNA-A (2.75 eV) were higher than that of a DOB2-DABNA-B-NP (2.60 eV). This can be attributed to the different structural torsion between DOB2-DABNA-A and DOB2-DABNA-B-NP, which affects the effectiveness of π -conjugation as discussed in the main text (Figure 1). Second, the S-T gaps of DOB2-DABNA-A and DOB2-DABNA-B-NP are smaller than DABNA-1 because of the charge delocalization through the π -extension. This trend is rationalized by theoretical calculation (Ref. 30, *Nat. Commun.* 2019, 10, 597.) and also observed in *v*-DABNA (Ref. 10). We suppose these two effects have caused the T₁ energies of DOB2-DABNA-A/B to be higher than that of DABNA-1. To clarify these points, we have revised the main text as follows.

(page 6)

Original: Based on the peak top of the fluorescence and phosphorescence spectra at 77 K, the S₁ and T₁ energies were determined as 2.745 and 2.742 eV, respectively. The estimated ΔE_{ST} value from the spectral data is extremely low (3.6 meV), confirming the potential of MR-TADF emitters.

Revised: Based on the peak top of the fluorescence and phosphorescence spectra at 77 K, the S₁ and T₁ energies were determined as 2.745 and 2.742 eV, respectively. The S₁ energy was not so different from DABNA-1 (2.71 eV) because the two nitrogen or two boron atoms in the meta-position are not effectively conjugated and change the orbital energy little, which is the general trend in MR-TADF materials.[10] In contrast, the T₁ energy of DOB2-DABNA-A was much higher than that of DABNA-1 (2.54 eV) (Figure S5). This originates from the significant decrease in ΔE_{ST} value (3.6 meV for DOB2-DABNA-A and 170 meV for DABNA-1, respectively), which can be attributed to the charge delocalization through the π -extension.[30]

Figure r1. S₁ and T₁ energies estimated from fluorescence and phosphorescence spectra at 77 K in 1 wt%-doped PMMA films.

[Comment #2]

Compared with DOB2-DABNA-A, the reference compound DOB2-DABNA-B has two different substituents except for the position where DOBNA merged with DABNA-1. However, from the theoretical calculation, the diphenylamine substituent of DOB2-DABNA-B participates in the frontier orbital distribution, so DOB2-DABNA-B is not a suitable reference. It is recommended that the authors synthesize model-B reference compounds with identical substituents and study the corresponding photophysical properties.

Response: Thank you for your constructive comment. We agree that the effect of diphenylamine group should be investigated for accurate comparison. For the comparison, we have synthesized DOB2-DABNA-A-NP that is the model-A derivative possessing a diphenylamine group instead of the model-B derivative with *tert*-butyl group, which will take some more time for synthesis. Compared to DOB2-DABNA-A, DOB2-DABNA-A-NP showed slightly shorter λ_{\max} (446 nm) and similar k_{RISC} value ($1.2 \times 10^6 \text{ s}^{-1}$ for DOB2-DABNA-A-NP). These data indicate that the diphenylamine group affects excitation energy but does not affect RISC process significantly. To verify this, we conducted NTO calculations for DOB2-DABNA-A and DOB2-DABNA-A-NP and confirmed that substituents at the central phenyl ring had little contribution to the S_0 - S_1 transition. We have added the NTOs to Figure S2 in the SI. Since DOB2-DABNA-A-NP showed lower PLQY ($\Phi = 0.76$) and broader emission spectrum (FWHM = 29 nm) than DOB2-DABNA-A, probably due to aggregation, we did not fabricate an OLED using DOB2-DABNA-A-NP. To clarify these issues, we have added the following discussion and figures to the manuscript.

(page 7)

Added: To evaluate the effect of substituents at the central phenyl ring, the photophysical properties of DOB2-DABNA-A and DOB2-DABNA-A-NP were compared. DOB2-DABNA-A-NP showed slightly shorter λ_{\max} (446 nm, Figure 3c) and similar k_{RISC} ($1.2 \times 10^6 \text{ s}^{-1}$, Figure 3d). These data indicate that the diphenylamine group affects excitation energy but does not significantly affect RISC process. To verify this, we conducted NTO calculations for DOB2-DABNA-A and DOB2-DABNA-A-NP and confirmed that substituents at the central phenyl ring had little contribution to the S_0 - S_1 transition (Figure S2). DOB2-DABNA-A-NP showed lower ($\Phi = 0.76$) PLQY and broader emission spectrum (FWHM = 29 nm) than DOB2-DABNA-A, probably due to aggregation in the PMMA film.

(Figure 3)

Original:

Figure 3. Photophysical properties of DOB2-DABNA-A (a,b) and DOB2-DABNA-B (c,d) in poly(methyl methacrylate) (PMMA;1-wt%-doped films). (a,c) Fluorescence spectra at 300 K (blue) and 77 K with (green) and without (red) a delay time of 25 ms. (b,d) Transient photoluminescence decay curves at 300 K and their relevant parameters. The red curves represent the single exponential fitting data (background = 2 or 3).

Revised:

Figure 3. Photophysical properties of **DOB2-DABNA-A** (a,b), **DOB2-DABNA-A-NP** (c,d), and **DOB2-DABNA-B-NP** (e,f) in 1 wt%-doped PMMA films. (a,c,e) Fluorescence spectra at 300 K (blue) and 77 K with (green) and without (red) a delay time of 25 ms. (b,d,f) Transient photoluminescence decay curves at 300 K and their relevant parameters. The red curves represent the single exponential fitting data (background = 1–3).

(Figure S2)

Added:

Figure S2. Natural transition orbitals in the $S_0 \rightarrow S_1$ transition of **DOB2-DABNA-A**, **DOB2-DABNA-A-NP**, and **DOB2-DABNA-B-NP** calculated at the B3LYP/6-31G(d) level of theory (isovalue = 0.02).

Relatedly, we have added a scheme for **DOB2-DABNA-A-NP** consisting of Buchwald–Hartwig coupling and one-shot borylation. To unify synthetic procedures for three compounds, we also conducted synthesis of **DOB2-DABNA-A** by one-shot borylation and added the results in Figure 2 as shown below. Accordingly, the name of the model-B-based compound was changed from **DOB2-DABNA-B** to **DOB2-DABNA-B-NP** to clarify the existence of a diphenylamine group. Detailed explanations of the synthesis were also added.

(Figure 2)
Original:

Figure 2. Synthesis of **DOB2-DABNA-A** and **DOB2-DABNA-B**.

Revised:

Figure 2. Synthesis of **DOB2-DABNA-A-NP**, **DOB2-DABNA-A**, and **DOB2-DABNA-B-NP**.

(page 5)

Added:

Motivated by the computational results described above, a new MR-TADF blue emitter, **DOB2-DABNA-A-NP**, was designed based on **model-A**. A diphenylamine group was introduced to the central benzene ring to enhance the nucleophilicity. The synthesis of **DOB2-DABNA-A-NP** is shown in Figure 2: Buchwald–Hartwig coupling between 3,5-dichloro-*N,N*-diphenylaniline and **DOBNA-NHPh** afforded intermediate **1**. In the presence of boron triiodide and 2,6-di-*tert*-butylpyridine, the following borylation of **1** consisting of

intermolecular and intramolecular electrophilic C–H borylation [60–62] smoothly took place at room temperature. Since we observed over borylation at the peripheral positions (not identified), the crude product was treated with acetic acid to remove the boryl groups to afford **DOB2-DABNA-A-NP** in 42% yield. This borylation method, known as one-shot borylation,[63–66] was also applicable for synthesizing **DOB2-DABNA-A**, which possesses a *tert*-butyl group instead of a diphenylamine group to suppress unfavorable intermolecular borylation at the central benzene ring. In addition, an *m*-xylyl group (denoted Xyl) was adopted as an *N*-attached aryl group to suppress the over borylation. As a result, **DOB2-DABNA-A** was selectively obtained in 40% yield without any treatment with acetic acid. As a reference compound based on **model-B**, we also synthesized **DOB2-DABNA-B-NP** possessing a diphenylamine group. To suppress undesired intramolecular borylation on the *N*-attached aryl groups, *o*-tolyl groups were used instead of phenyl and *m*-xylyl groups. Since the borylation using boron triiodide and 2,6-di-*tert*-butylpyridine gave a complex mixture, we chose boron tribromide as a milder borylating reagent. As a result, the reaction took place under reflux condition of chlorobenzene to give **DOB2-DABNA-B-NP** in 41% yield upon removal of the borylated groups at the peripheral positions by treatment with acetic acid.

[Comment #3]

The nature of the emitter is undoubtedly essential to the device's performance. Still, combined with the research work of this group and all others on MR-TADF's, the smaller efficiency roll-off of the device in this work seems to be more derived from the selection of the host material, ETL material, and blocking layer between them. The author should add relevant explanations in the text. Besides, have the authors studied the EL properties of devices based on the commonly used host materials (mCBP/TSPO1) and/or ETL (TPBi/Bphen/TmPyPB)?

Response: We appreciate the insightful feedback for our manuscript. We fabricated an OLED device using **DOB2-DABNA-A** as an emitter with a different host. Because mCBP and TSPO1 have much larger I_p and much smaller E_a than the emitter (Table r1), the emitter can trap both hole and electron to reduce the charge recombination efficiency (*Angew. Chem. Int. Ed.* **2023**, *62*, 20227512). Instead of mCBP and TSPO1, we used SiTrzCz2 possessing close E_a to the emitter as a host material (use in deep-blue OLEDs: *Sci. Adv.*, 2022, **8**, eabq1641, *Adv. Mater.* 2023, **35**, 2210794, *Adv. Sci.* 2023, *10*, 2301112). The resulting EQE_{max} was 22.3%, and EQE@1000 cd m⁻² was 17.0%. Considering the lack of TADF property of SiTrzCz2, this result clearly indicates that large k_{RISC} of **DOB2-DABNA-A** is the principal reason for the small efficiency roll-off. Note that the EQE values with SiTrzCz2 are slightly lower than the device with **DABNA-Tol** host (24.1% and 21.6%) probably because of lower charge recombination efficiency caused by larger I_p (6.0 eV, 5.73 eV) and slightly smaller E_a (2.8 eV, 2.99 eV) of SiTrzCz2 than **DOB2-DABNA-A**.

Table r1. Ionization Potential (I_p), Optical Band Gap (E_g), and Electron Affinity (E_a) of emitters and host materials.

compound	HOMO [eV]	I_p [eV]	E_g [eV]	E_a [eV]
DABNA-1 ^a	-4.74	5.58	2.67	2.91
DOB2-DABNA-A	-4.89	5.73 ^b	2.74 ^c	2.99 ^d
DOB2-DABNA-B-NP	-5.03	5.87 ^b	2.62 ^c	3.25 ^d
DOBNA-Tol		6.2		3.2
mCBP ^e		6.1		2.6
SiTrzCz2 ^f		6.0		2.8
TSPO1 ^g		6.8		2.5

^a*Adv. Mater.* **2016**, *28*, 2777. ^bEstimated from I_p of **DABNA-1** and HOMO energy levels. ^cEstimated from the onset wavelength of the UV-vis absorption spectrum in 1 wt% PMMA film. ^dCalculated from I_p and E_g . ^e*Angew. Chem. Int. Ed.* **2023**, *62*, e202217512. ^f*Sci. Adv.*, **2022**, *8*, eabq1641. ^g*Adv. Mater.* **2011**, *23*, 1436.

Figure r2. Characteristics of fabricated OLED device using **DOB2-DABNA-A** as an emitter in **SiTrzCz2** host. a) Device structure, ionization potentials (I_p), and electron affinities (E_a ; in eV) for each component. b) Normalized EL spectra of the devices in operation. c) Commission Internationale de l'Éclairage (CIE) (x,y) coordinates. d) Current density (blue) and luminance (green) vs. driving voltage. e) EQE vs. luminance.

Furthermore, we measured the photophysical data of **DOBNA-Tol** neat film and 1 wt%-doped **DOBNA-Tol** films to evaluate the effect of the host matrix in the OLED device (Figures S6 and S8). The k_{RISC} values are in the following order: **DOBNA-Tol** neat film ($8.8 \times 10^4 \text{ s}^{-1}$) < 1 wt%-doped **DOBNA-Tol** film ($6.0 \times 10^5 \text{ s}^{-1}$ for **DOB2-DABNA-A** and $2.0 \times 10^5 \text{ s}^{-1}$ for **DOB2-DABNA-B-NP**) < 1 wt%-doped PMMA film ($1.1 \times 10^6 \text{ s}^{-1}$ for **DOB2-DABNA-A** and $3.8 \times 10^5 \text{ s}^{-1}$ for **DOB2-DABNA-B-NP**), suggesting the RISC process in PMMA is faster than in **DOBNA-Tol**. These data also support our claim that the high k_{RISC} values of emitters suppressed the efficiency roll-off to realize the highest EQE (21.6%) at the practical luminance (1000 cd m^{-2}). We added the following comment and figures in the manuscript.

(page 10)

Added: Furthermore, the k_{RISC} values of a neat film of **DOBNA-Tol** ($8.8 \times 10^4 \text{ s}^{-1}$) and the 1 wt%-doped **DOBNA-Tol** films ($6.0 \times 10^5 \text{ s}^{-1}$ for **DOB2-DABNA-A** and $2.0 \times 10^5 \text{ s}^{-1}$ for **DOB2-DABNA-B-NP**) were smaller than the 1 wt%-doped PMMA films ($1.1 \times 10^6 \text{ s}^{-1}$ and $3.8 \times 10^5 \text{ s}^{-1}$) (Figures S6 and S8). This allows us to rule out the possibility of a sensitizing effect of **DOBNA-Tol**.

(Figure S6)

Added:

Figure S6. Photophysical properties of **DOB2-DABNA-A** (a,b) and **DOB2-DABNA-B-NP** (c,d) in 1 wt%-doped **DOBNA-Tol** films. (a,c) Fluorescence spectra at 300 K (blue) and 77 K with (green) and without (red) a delay time of 25 ms. (b,d) Transient photoluminescence (PL) decay curves at 300 K and their relevant parameters. The red curves represent the single exponential fitting data (background = 3.32–5.32).

(Figure S8)

Added:

Figure S8. Photophysical properties of a neat film of **DOBNA-Tol**. (a) Fluorescence spectra at 300 K (blue) and 77 K with (green) and without (red) a delay time of 25 ms. (b,d) Transient photoluminescence (PL) decay curves at 300 K and their relevant parameters. The red curves represent the single exponential fitting data (background = 4).

[Comment #4]

As stated by the authors, DOB2-DABNA-B exhibited longer wavelength emissions than DOB2-DABNA-A, consistent with theoretical predictions. However, compared with DABNA-1, it is inconsistent with the theoretical prediction. Can the author give a reasonable explanation?

Response: Thank you for pointing it out. It is supposed to be an intrinsic error of DH-TD-DFT calculations. The CCSD and CC2 methods can suppress such errors, but they are not applicable to model-A/B, which consists of a large number of basis functions.

[Comment #5]

In Page 1 (line 21), it should be DOB2-DABNA-A instead of DOB2-DABNA. Besides, the efficiency roll-off of the device should be 6.1% (1.1%/18%) instead of 1.1% (line 23, page 8).

Response: Thank you for your careful reviewing. We corrected errors the reviewer pointed out. Because we recollected the OLED data, the efficiency roll-off was revised as 10.4% (2.5%/24.1%).

Reviewer #2 (Remarks to the Author):

The manuscript nat comm 438714_0 “Highly Efficient Multi-Resonance Thermally Activated Delayed Fluorescence Material Toward a BT.2020 Deep-Blue Emitter” describes a new MR-TADF emitters synthesis, photophysical and OLED device analysis.

Generally, the manuscript is correctly written and presents interesting results. I admire Prof. Hatakeyama's work, but unfortunately, in my opinion, this work has substantial errors that are impossible to correct without losing the main point and should be rejected.

I believe the photophysical analysis is misleading, and the OLED device is wrong (probably the NIST calibration is missing) because those materials couldn't get such high EQE values. Many authors wrongly analyse the PLQY measurement and associate all the efficiency to the DF process, which is untrue. Pure fluorescence emitter can also have PLQY 100%, but we cannot get more than 5% EQE.

Here authors present emitters with high PLQY values, but the triplet contribution is very low (table 1 - from 11% up to 0.42%). The DF/PF or ϕ_{DF}/ϕ_F values are very low <0.74, which tells us that we even cannot use the equations mentioned by the authors because not all assumptions were fulfilled. To correctly calculate the kinetic parameters, we need strong TADF emitters with $\phi_{DF}/\phi_F > 4$ (10.1088/2050-6120/aa537e).

In that case, it would be impossible to get the EQE of the devices based on those compounds with values above 8%.

I think a broader photophysical analysis should be done, and the calibration of the photoluminescence spectrometer and OLED system should be checked.

Response: The requirement for ϕ_{DF}/ϕ_F value which the reviewer pointed out is derived from the following equation:

$$\Phi_F = \Phi_{PF} + \Phi_{DF} = \sum_{i=0}^n \Phi_{PF} (\Phi_{ISC} \Phi_{RISC})^i = \Phi_{PF} \frac{1}{1 - \Phi_{ISC} \Phi_{RISC}}$$

This assumes photoluminescence (PL). In the PL process, excited molecules from the S_0 state at first occupy the S_1 state because it is a spin-allowed transition. If the exciton soon relaxes to the S_0 state, prompt fluorescence (PF) can be observed within a nano-second scale (red arrows in Figure r3, left). On the other hand, the delayed fluorescence (DF) occurs via the T_1 state (blue arrows in Figure r3, left). Due to the spin-forbidden character in the $S_1 \rightarrow T_1$ transition, the DF possesses a micro-second lifetime. The term $\Phi_{ISC} \Phi_{RISC}$ in the above equation represents the $S_1 \rightarrow T_1 \rightarrow S_1$ process.

In the PL process, the Φ_{PF}/Φ_{DF} ratio roughly corresponds to the k_r/k_{ISC} ratio, assuming that there is no nonradiative decay process from the T_1 state. Typical MR-TADF materials have higher k_r and lower k_{ISC} values than donor-acceptor (D-A) type TADF ones, and therefore Φ_{DF} is not so high (<0.2).[Refs.9–15] However, this does not suppress the EQE values because the pathway up to fluorescence is completely different in OLED devices. The EQE values are related to electroluminescence (EL), which starts from 25% S_1 and 75% T_1 excitons immediately after charge recombination (Figure r3, center). The 75% T_1 excitons are the main component of DF through a $T_1 \rightarrow S_1 \rightarrow S_0$ pathway, responsible for higher EQEs of TADF-OLEDs (20~30%) than fluorescence OLEDs (5~8%). Although the DF also involves a $S_1 \rightarrow T_1 \rightarrow S_1 \rightarrow S_0$ pathway, this is a very minor component because of the large k_r/k_{ISC} ratio of MR-TADF materials, that is, the above equation for the PL process cannot be applied for the EL process. On the other hand, k_{RISC} is the critical parameter for determining EL performance. The k_{RISC} values of MR-TADF materials are typically $10^4 - 10^5 \text{ s}^{-1}$, which is sufficient to obtain EQE_{max} of 20–30% at the low current density.[Refs.9–15] However, the k_{RISC} of $10^4 - 10^5 \text{ s}^{-1}$ is insufficient to suppress the efficiency roll-off at the high current density (luminance) where triplet-triplet annihilation (TTA) or triplet-polaron annihilation (TPA) cause non-radiative processes from the T_1 state. Therefore, we have developed **DOB2-DABNA-A** showing the highest k_{RISC} ($1.1 \times 10^6 \text{ s}^{-1}$) among the deep-blue emissive materials to suppress the roll-off and achieve the highest EQE at 1000 cd m^{-2} (21.6 %).

Because the paper suggested by the reviewer (10.1088/2050-6120/aa537e) contains beneficial information about the PL process of TADF materials, we added it as reference 6. We thank the reviewer's suggestion.

Figure r3. Schematic illustration of PL and EL processes.

Reviewer #3 (Remarks to the Author):

See attachment.

Response: We appreciate the reviewer's high evaluation of our manuscript.

Reviewer #4 (Remarks to the Author):

Prof. Hatakeyama et al. have presented a study on an ultrapure deep blue MR-TADF emitter, which exhibits a fast krisc of over 10^6 . The OLED demonstrates narrowband deep blue EL with CIE coordinates of (0.14, 0.05) and a mild efficiency roll-off at 1000 nits, which is an important target of OLEDs. I thus believe the current results are supportive to be a candidate for Nat. Commun. paper. However, the current data and discussions are far from enough. A major revision is needed.

[Comment #1]

The authors have noted that both emitters have small ΔE_{ST} values, but there is a significant difference in their krisc values. The SOC between S1 and T1 is not particularly high, so it is important to explore why DOB2-DABNA-A can achieve a krisc of over 10^6 s⁻¹. What is the contribution of higher lying triplet states. The current discussion is insufficient, and more theoretical estimations and discussions are required.

Response:

Thank you for your valuable comment. Assuming thermal equilibrium between the T₁ and T₂ states, the T₂/T₁ ratios are estimated to be 5.2×10^{-5} and 3.1×10^{-8} from $\Delta E_{T_1-T_2}$ values of 0.255 and 0.447 eV, respectively (Figure 1). Thus, the contribution of T₂ excited state is expected to be negligible. Moreover, the k_{RISC} value from the T₁ state was estimated according to the literature (7). Based on the semi-classical Marcus equation, the k_{RISC} value can be denoted as:

$$k_{RISC} = \frac{2\pi}{\hbar g} \frac{V_{SOC}^2}{\sqrt{4\pi k_B T \lambda}} \exp\left(-\frac{(\lambda + \Delta G^\circ)^2}{4\lambda k_B T}\right) \quad (i)$$

where \hbar , g , V_{SOC} , k_B , T , λ , and ΔG° represent the Dirac's constant, degeneracy of the initial state ($g = 3$ for calculating RISC rate), spin-orbit coupling, Boltzmann constant, temperature, reorganization energy, and change in free energy between the energies of the respective initial triplet and final singlet states, respectively. λ is assumed to be 0.10 eV because the structural relaxation of MR-TADF emitters is generally small because of their rigid framework.

Based on equation (i), the k_{RISC} value between S₁-T₁ states was estimated (Table S3). The k_{RISC} values estimated by using calculated $\Delta E_{S_1-T_1}$ confirmed the superiority of model-A to model-B, corresponding to the experimental result. Moreover, using the experimental $\Delta E_{S_1-T_1}$ improved the estimated values to get close to the accurate ones. These results suggest the decreased $\Delta E_{S_1-T_1}$ by merging two DOBNA units plays a key role in accelerating k_{RISC}. We added the following comments to the manuscript.

(page 3)

Original: The enlarged π -conjugation decreased both the $\Delta E_{S_1-T_1}$ (23 meV for **model-A**, 80 meV for **model-B**, and 239 meV for **DABNA-1**) and $\Delta E_{S_1-T_2}$ (231 meV for **model-A**, 367 meV for **model-B**, and 586 meV for **DABNA-1**) values, which was expected to improve the TADF properties. Next, the potential of the two models as a BT.2020 blue emitter was investigated. The spin-orbit coupling (SOC) matrix elements ($\langle S_n | \hat{H}_{SOC} | T_n \rangle$) calculated at an M062X/TZP level of theory indicate that **model-A** has a large SOC (S₁-T₁: 0.059 cm⁻¹, S₁-T₂: 0.895 cm⁻¹) compared to **model-B** (S₁-T₁: 0.035 cm⁻¹, S₁-T₂: 0.206 cm⁻¹), benefiting from the more helical structure. [52]

Revised:

The enlarged π -conjugation decreased both the $\Delta E_{S_1-T_1}$ (23 meV for **model-A**, 80 meV for **model-B**, and 239 meV for **DABNA-1**) and $\Delta E_{S_1-T_2}$ (231 meV for **model-A**, 367 meV for **model-B**, and 586 meV for **DABNA-1**) values. The spin-orbit coupling (SOC) matrix elements ($\langle S_n | \hat{H}_{SOC} | T_n \rangle$) calculated at an M062X/TZP level of theory indicate that **model-A** has a large SOC (S₁-T₁: 0.059 cm⁻¹, S₁-T₂: 0.895 cm⁻¹) compared to **model-B** (S₁-T₁: 0.035 cm⁻¹, S₁-T₂: 0.206 cm⁻¹), benefiting from the more helical structure. [53] Although the contribution of T₂ to the RISC process is expected to be very small due to the large $\Delta E_{S_1-T_2}$ values, the small $\Delta E_{S_1-T_1}$ and the larger SOC for S₁-T₁ will improve the TADF properties.

(page 8)

Added: The higher k_{RISC} in **DOB2-DABNA-A** derivatives than **DOB2-DABNA-B-NP** could be reproduced by semi-classical Marcus equation using computational and experimental parameters (Table S3).

(Table S3)

Added: **Table S3.** Summary of Transition Energies, SOC Matrix Elements, and Calculated Photophysical Data of **model-A**, **model-B**, and **DABNA-1**.

compound	calculation					hybrid		experiment
	$\text{SOC}_{T_1-S_1}^a$ [cm ⁻¹]	E_{S_1} [eV]	E_{T_1} [eV]	$\Delta E_{S_1-T_1}^b$ [meV]	$k_{\text{RISC}}^{c,d}$ [s ⁻¹]	$\Delta E_{S_1-T_1}^e$ [meV]	$k_{\text{RISC}}^{c,f}$ [s ⁻¹]	k_{RISC}^e [s ⁻¹]
model-A	0.059	2.682	2.658	23	3.9×10^5	3.6^g	8.3×10^5	$1.1 \times 10^6^g$
						18^h	4.7×10^5	$1.2 \times 10^6^h$
model-B	0.035	2.529	2.449	80	1.5×10^4	6.5	2.6×10^5	3.8×10^5
DABNA-1	0.070	2.732	2.492	239	1.3×10^2	170	1.9×10^3	4.8×10^3

^aSpin-orbit coupling between S₁ and T₁ states. ^bEnergy gap between S₁ and T₁ states estimated from the calculation. ^c $k = 2\pi/hg(V_{\text{SOC}}^2/4\pi k_{\text{B}}T\lambda)\exp(-\Delta E_{(S_1-T_1)}/k_{\text{B}}T)$. λ is assumed to be 0.10 eV. ^dEstimated by using the calculated $\Delta E_{S_1-T_1}$ value. ^eSpectroscopic data in PMMA films (1 wt%). ^fEstimated by using the experimental $\Delta E_{S_1-T_1}$ value. ^gValues of **DOB2-DABNA-A**. ^hValues of **DOB2-DABNA-A-NP**.

[Comment #2]

The authors should provide an explanation for their choice of PMMA as the dilute host. It is important to consider the potential aggregation of the MR emitters during spin-coating, as it can impact the photophysics. Please note the facts that the FWHM of DOB2-DABNA-A in PL is even boarder than that in EL spectrum, and PL of DOB2-DABNA-B at R.T is different from that at 77 K (Figure 3c), which are abnormal.

Response: Thank you for your insightful comment. In the original manuscript, we adopted PMMA matrix to evaluate only the effect of molecular structure (model-A vs model-B), eliminating interactions with host matrices. Since we agree that photophysical properties in the host material should be measured to connect PL and EL properties, we added the photophysical measurement data of 1 wt%-doped **DOBNA-ToI** film (Figure S6). The FWHM values of **DOB2-DABNA-A** and **DOB2-DABNA-B-NP** were 27/28 nm (1 wt% PMMA), 24/23 nm (1 wt% **DOBNA-ToI**), and 24/23 nm (OLED device). The PL in the host material is consistent with the EL spectrum, especially in terms of FWHM. To elucidate the origin of broadening in PMMA, we also measured a PL spectrum of **DOB2-DABNA-A** in 0.1 wt%-doped PMMA film (Figure S7). The fluorescent peak was 444 nm, and the FWHM value was 25 nm (451 nm and 27 nm for 1 wt%-doped PMMA film, respectively). The red-shifted and broadened spectrum in the 1 wt%-doped film suggest aggregation of **DOB2-DABNA-A**. We added the following description in the manuscript.

(page 9)

Added: To evaluate the potential as OLED emitters, their photophysical properties in the 1 wt%-doped **DOBNA-ToI** host material^[64] were measured (Figure S6). The fluorescence peaks were at the same position as PMMA films. The FWHM values of **DOB2-DABNA-A** and **DOB2-DABNA-B-NP** in **DOBNA-ToI** were 24 nm and 23 nm, which was smaller than those in PMMA (27 nm and 28 nm). This can be attributed to aggregation of **DOB2-DABNA-A** in the 1 wt%-doped PMMA films, as also suggested by the decreased FWHM value of 0.1 wt%-doped PMMA film (23 nm, Figure S7).

(Figure S6)

Added:

Figure S6. Photophysical properties of **DOB2-DABNA-A** (a,b) and **DOB2-DABNA-B-NP** (c,d) in 1 wt%-doped **DOBNA-Tol** films. (a,c) Fluorescence spectra at 300 K (blue) and 77 K with (green) and without (red) a delay time of 25 ms. (b,d) Transient photoluminescence (PL) decay curves at 300 K and their relevant parameters. The red curves represent the single exponential fitting data (background = 3.32–5.32).

(Figure S7)

Added:

Figure S7. Fluorescence spectrum of **DOB2-DABNA-A** in 0.1 wt%-doped PMMA film at 300 K.

Thank you for pointing out the red-shift in **DOB2-DABNA-B-NP** from 300 K to 77 K. Similar spectral shifts have been often observed for MR-TADF materials in toluene (Ref.10, *Adv. Electron. Mater.* **2021**, 7, 2001090), PMMA (Refs. 13, 15), and **DPEPO** (*J. Mater. Chem. C.* **2022**, 10, 11855.); however, they have not been discussed and elucidated so far. To achieve a deeper insight, we have measured PL spectra of 1 wt%-doped **DOBNA-Tol** film at 77 and 300 K (Figure S6) in which the red-shift was not observed. Considering these results, stronger interaction between the emitter and PMMA at 77 K than 300 K may cause the spectral shift. To avoid non-reliable speculation, we added the following comments in the manuscript.

(page 10)

Added: Note that the unusual λ_{\max} shift between 300 K and 77 K observed for **DOB2-DABNA-B-NP** in the PMMA film (Figure 3e) was not reproduced in the **DOBNA-Tol** film.

[Comment #3]

It is recommended to measure temperature-dependent transient PL decays and estimate the ΔE_{ST} using an Arrhenius plot based on the collected data.

Response: Thank you for your valuable comment. We prepared Arrhenius plots of $k_{\text{RISC}}T^{0.5}$ and $k_{\text{ISC}}T^{0.5}$ versus $1/T$ for synthesized compounds (Figure S9). The activation energies for the RISC ($\Delta E_{\text{a}}^{\text{RISC}}$) and ISC ($\Delta E_{\text{a}}^{\text{ISC}}$) processes were estimated to be 64 and 27 meV (for **DOB2-DABNA-A**), 70 and 26 meV (for **DOB2-DABNA-A-NP**), and 43 and 52 meV (for **DOB2-DABNA-B-NP**), respectively. The reversal of the activation energies in **DOB2-DABNA-B-NP** ($\Delta E_{\text{a}}^{\text{RISC}} < \Delta E_{\text{a}}^{\text{ISC}}$) can be attributed to the dark state in equilibrium with the T_1 state. This reduces population of the T_1 state at higher temperature to decrease k_{RISC} and the slope ($\Delta E_{\text{a}}^{\text{RISC}}$) of the fitting line from the original one. We have added the following comments in the manuscript.

(page 8)

Added: Transient spectra of all three compounds at different temperatures showed that the TADF component vs. the prompt one decreased at lower temperatures (Figure S9). From the Arrhenius plots of $k_{\text{RISC}}T^{0.5}$ and $k_{\text{ISC}}T^{0.5}$ versus $1/T$, the activation energies for the RISC ($\Delta E_{\text{a}}^{\text{RISC}}$) and ISC ($\Delta E_{\text{a}}^{\text{ISC}}$) processes were estimated to be 64 and 27 meV (for **DOB2-DABNA-A**), 70 and 26 meV (for **DOB2-DABNA-A-NP**), and 43 and 52 meV (for **DOB2-DABNA-B-NP**), respectively. The reversal of the activation energies in **DOB2-DABNA-B-NP** can be attributed to the dark state in equilibrium with the T_1 state, which may reduce population of the T_1 state at higher temperature to decrease k_{RISC} and the slope ($\Delta E_{\text{a}}^{\text{RISC}}$) of the fitting line.

(Figure S9)

Added:

Figure S9. Photophysical properties of (a,b) **DOB2-DABNA-A**, (c,d) **DOB2-DABNA-A-NP**, and (e,f) **DOB2-DABNA-B-NP** in PMMA (1 wt%-doped film). (a,c,e) Transient decay spectra at 150 (purple), 175 (indigo), 200 (blue), 225 (green), 250 (yellow), 275 K (orange), and 300 K (red). (b,d,f) Arrhenius plots of $k_{\text{RISC}}T^{0.5}$ (red) and $k_{\text{ISC}}T^{0.5}$ (blue) vs. $1/T$.

[Comment #4]

It would be valuable to investigate whether DOBNA-Tol acts as a TADF host and plays a role as a sensitizer for the terminal emitter.

Response: We added photophysical data of DOBNA-Tol neat film (Figure S8). The estimated k_{RISC} value was $8.8 \times 10^4 \text{ s}^{-1}$, which was smaller than the terminal emitters in PMMA matrix (3.8×10^5 – $1.2 \times 10^6 \text{ s}^{-1}$). Furthermore, the k_{RISC} value of DOB2-DABNA-A in DOBNA-Tol (1 wt%) was $6.0 \times 10^5 \text{ s}^{-1}$, which is also smaller than that in PMMA (1 wt%) ($1.1 \times 10^6 \text{ s}^{-1}$). Therefore, DOBNA-Tol acts as a TADF host rather than a sensitizer unit.

(page 10)

Added: Furthermore, the k_{RISC} values of DOBNA-Tol neat film ($8.8 \times 10^4 \text{ s}^{-1}$) and the 1 wt%-doped DOBNA-Tol films ($6.0 \times 10^5 \text{ s}^{-1}$ for DOB2-DABNA-A and $2.0 \times 10^5 \text{ s}^{-1}$ for DOB2-DABNA-B-NP) were smaller than the 1 wt%-doped PMMA films ($1.1 \times 10^6 \text{ s}^{-1}$ and $3.8 \times 10^5 \text{ s}^{-1}$) (Figures S6 and S8). This allows us to rule out the possibility of a sensitizing effect of DOBNA-Tol.

(Figure S8)

Added:

Figure S8. Photophysical properties of a neat film of DOBNA-Tol. (a) Fluorescence spectra at 300 K (blue) and 77 K with (green) and without (red) a delay time of 25 ms. (b,d) Transient photoluminescence (PL) decay curves at 300 K and their relevant parameters. The red curves represent the single exponential fitting data (background = 4).

[Comment #5]

The measurement of device operational lifetime is crucial, particularly for deep blue OLED emitters. This parameter should be included and discussed in the study.

Response: Thank you for your important suggestion. To evaluate the operational lifetime accurately, we refabricated the OLED device with the same layer structure using our new vacuum deposition equipment. As a result, the half-lifetime (LT_{50}) of the DOB2-DABNA-A-based device under this condition was 52 h, which is over ten times longer than the previous deep-blue MR-TADF emitter-based OLEDs. The lifetime data have been added in the main text and Table S4 and Figure S14. Consequently, the OLED performance was improved from one reported in the original manuscript (*i.e.* EQE_{max} from 18.0% to 24.1%, EQE_{1000} from 16.9% to 21.6%). This is probably because of contamination and/or poor concentration control on the vapor deposition process of our old equipment, which was used to fabricate the device reported in the original manuscript. Accordingly, we have revised the data in Figure 4 and Tables S4 and S5 and the related discussion in the main text.

(page 11)

Added: Moreover, the half-lifetime (LT_{50}) of the DOB2-DABNA-A-based device with an initial luminance of 100 cd m^{-2} was 52 h, which is over ten times longer than the previous deep-blue MR-TADF emitter-based OLEDs (Figure S14 and Table S4). [41,43,44,50]

(Figure 4)
Original:

Figure 4. Characteristics of fabricated OLED device using **DOB2-DABNA-A** as an emitter. a) Device structure, ionization potentials (I_p), and electron affinities (E_a ; in eV) for each component. The I_p and E_a of **DOB2-DABNA-A** were estimated from those of **DABNA-1** and their HOMO/LUMO energy levels. b) Normalized EL spectra of the devices in operation. Inset: electroluminescence of the device. c) Commission Internationale de l'Éclairage (CIE) (x,y) coordinates of (blue) **DOB2-DABNA-A** and (red) BT.2020 blue emitter d) Current density (blue) and luminance (green) vs. driving voltage. e) EQE vs. luminance. f) Comparison of EQE values at 1000 cd m^{-2} for reported deep-blue OLEDs with $\text{CIE}_y < 0.1$. g) Comparison of k_{RISC} values at 1000 cd m^{-2} for reported deep-blue OLEDs with $\text{CIE}_y < 0.1$.

Revised:

Figure 4. Characteristics of fabricated OLED device using **DOB2-DABNA-A** (blue) and **DOB2-DABNA-B-NP** (light blue) as an emitter. a) Device structure, ionization potentials (I_p), and electron affinities (E_a ; in eV) for each component. The I_p and E_a of emitters were estimated from those of **DABNA-1** and their HOMO/LUMO energy levels. b) Normalized EL spectra of the devices in operation. Inset: electroluminescence of the device. c) Commission Internationale de l'Éclairage (CIE) (x,y) coordinates. d) Current density (solid) and luminance (dashed) vs. driving voltage. e) EQE vs. luminance. f) Comparison of EQE values at 1000 cd m^{-2} for reported deep-blue OLEDs with $\text{CIE}_y < 0.1$. g) Comparison of k_{RISC} values at 1000 cd m^{-2} for reported deep-blue OLEDs with $\text{CIE}_y < 0.1$.

(Table S4)

Original:

Table S3. Summary of OLED Performance Employing Deep-Blue MR-TADF Emitter with CIE_y of ≤ 0.05 .

Emitter	λ_{EL}^a [nm]	FWHM ^b [nm]	CIE ^c (x, y)	V _{on} ^d [nm]	EQE ^e [%]	CE _{max} ^f [cd A ⁻¹]	PE _{max} ^g [lm W ⁻¹]
DOB2-DABNA-A	452	24	(0.14, 0.05)	3.5	18.0/18.0/16.9	9.7	8.8
BOBO-Z	445	18	(0.15, 0.04)	4.5	13.6/9.8/3.3	7.2	5
CzBO	448	30	(0.15, 0.05)	4.1	13.4/8.4/3.5	7.4	5.7
B-O-dpa	443	32	(0.15, 0.05)	3.8	16.3/2.2/-	8.3	-
BisICz	437	24	(0.16, 0.04)	-	6.5/(2.0)/-	2.9	2.7
tBisICz	445	22	(0.16, 0.05)	-	15.1/(3.0)/-	8.4	8.3
tPBisICz	452	21	(0.15, 0.05)	-	23.1/(5.0)/-	13.5	13.3
NOBNacene	409	37	(0.173, 0.055)	4.2	8.5/-/-	-	1.2
BIC-mCz	432	42	(0.16, 0.05)	3.3	19.4/(4.0)/-	-	9.7
mDBIC	431	42	(0.16, 0.05)	3.4	13.5/(3.0)/-	-	9.7
CZCO	432	35	(0.154, 0.047)	3.4	15.6/6.0/5.9	8.6	7.1
1B-DTACrs	440	30	(0.154, 0.049)	10	1.31/-/-	-	-
2B-DTACrs	447	26	(0.150, 0.044)	4	14.8/(10.0)/-	-	-
tDIDcz	401	17	(0.164, 0.055)	4.25	2.46/-/-	0.39	-
gm-lcz	412	36	(0.160, 0.034)	3.75	3.04/-/-	0.68	-
o-lcz	433	30	(0.165, 0.046)	3.5	3.01/-/-	1.28	-
MesB-DIDOBNA-N	402	21	(0.170, 0.049)	4.1	16.2/-/3.5	-	2.7

^aMaximum wavelength of EL spectrum. ^bFull width at half maximum. ^cCIE (x, y) coordinates. ^dTurn-on voltage at the luminescence of 1 cd m⁻². ^eExternal quantum efficiency of maximum/at 100/1000 cd m⁻². ^fMaximum current efficiency. ^gMaximum power efficiency.

Revised:

Table S4. Summary of OLED Performance Employing Deep-Blue MR-TADF Emitter with CIE_y of ≤ 0.05 .

Emitter	λ_{EL}^a [nm]	FWHM ^b [nm]	CIE ^c (x, y)	V _{on} ^d [nm]	EQE ^e [%]	CE _{max} ^f [cd A ⁻¹]	PE _{max} ^g [lm W ⁻¹]	LT50 ^h [h]	ref
DOB2-DABNA-A	452	24	(0.145, 0.049)	3.4	24.1/23.3/21.6/11.1	12.1	11.3	52	This work
BOBO-Z	445	18	(0.15, 0.04)	4.5	13.6/9.8/3.3/-	7.2	5.0	0.2 ⁱ 4.4 ^j	27
CzBO	448	30	(0.15, 0.05)	4.1	13.4/8.4/3.5/-	7.4	5.7	0.16	28
B-O-dpa	443	32	(0.15, 0.05)	3.8	16.3/2.2/-/-	8.3	-	(0.07) ^k	29
BisICz	437	24	(0.16, 0.04)	-	6.5/(2.0)/-/-	2.9	2.7	-	30
tBisICz	445	22	(0.16, 0.05)	-	15.1/(3.0)/-/-	8.4	8.3	-	30
tPBisICz	452	21	(0.15, 0.05)	-	23.1/(5.0)/-/-	13.5	13.3	-	30
NOBNacene	409	37	(0.173, 0.055)	4.2	8.5/-/-/-	-	-	-	31
BIC-mCz	432	42	(0.16, 0.05)	3.3	19.4/(4.0)/-/-	-	9.7	-	32
mDBIC	431	42	(0.16, 0.05)	3.4	13.5/(3.0)/-/-	-	9.7	-	32
CZCO	432	35	(0.154, 0.047)	3.4	15.6/6.0/5.9/-	8.6	7.1	-	33
1B-DTACrs	440	30	(0.154, 0.049)	10	1.31/-/-/-	-	-	-	34
2B-DTACrs	447	26	(0.150, 0.044)	4.0	14.8/10.1/-/-	-	-	-	34
tDIDcz	401	17	(0.164, 0.055)	4.25	2.46/-/-/-	0.39	-	-	35
gm-lcz	412	36	(0.160, 0.034)	3.75	3.04/-/-/-	0.68	-	-	35
o-lcz	433	40	(0.165, 0.046)	3.5	3.01/-/-/-	1.28	-	-	35
MesB-DIDOBNA-N	402	21	(0.170, 0.049)	4.1	16.2/3.5/-/-	-	2.7	<1.0	36

^aMaximum wavelength of EL spectrum. ^bFull width at half maximum. ^cCIE (x, y) coordinates. ^dTurn-on voltage at the luminescence of 1 cd m⁻². ^eExternal quantum efficiency of maximum/at 100/1000/10000 cd m⁻². ^fMaximum current efficiency. ^gMaximum power efficiency. ^hDevice half-lifetime with initial luminance of 100 cd m⁻². ⁱIn mCBP host. ^jIn mCBP-CN host. ^kInitial luminescence of 10 cd m⁻².

(Table S5)

We revised the OLED data of **DOB2-DABNA-A**.

(Figure S14)

Added:

Figure S14. Normalized luminance versus time characteristics of OLED device employing **DOB2-DABNA-A** (blue) and **DOB2-DABNA-B-NP** (light blue) with initial luminance at (a,b) 500 and (c,d) 100 cd m⁻². The LT50 value of **DOB2-DABNA-B-NP** was estimated by the equation: [LT50@100 cd m⁻²] = [LT50@500 cd m⁻²] × (500/100)ⁿ. n was estimated to be 1.39 by fitting the data at 500 cd m⁻² (b) to that at 100 cd m⁻² (d).

[Comment #6]

The EL performance of DOB2-DABNA-B should also be presented and discussed to provide a comprehensive analysis.

Response: We thank the reviewer for the constructive comment. We fabricated the OLED device by using **DOB2-DABNA-B** as an emitter. The data were merged to **DOB2-DABNA-A** and provided in Figures 4, S13, and S14. We also added a comprehensive discussion in the paragraph that describes the OLED data (page 10).

(Figure S13)

Added:

Figure S13. Current efficiency (solid) and power efficiency (dashed) versus luminance of OLED device fabricated with **DOB2-DABNA-A** (blue) and **DOB2-DABNA-B-NP** (light blue).

REVIEWERS' COMMENTS

Reviewer #1 (Remarks to the Author):

The authors have already solved the problems and confused existence. I think this revised manuscript can be accepted directly.

Reviewer #4 (Remarks to the Author):

The authors have well addressed the raised issues. I believe it can be accepted as is.